# Studying memorization of large language models using answers to Stack Overflow questions

**Laura Caspari**                                                    *laura.caspari@uni-passau.de*
*Faculty of Computer Science and Mathematics*
*University of Passau*
*Passau, Germany*

**Alexander Trautsch**                                         *alexander.trautsch@uni-passau.de*
*Faculty of Computer Science and Mathematics*
*University of Passau*
*Passau, Germany*

**Michael Granitzer**                                          *michael.granitzer@uni-passau.de*
*Faculty of Computer Science and Mathematics*
*University of Passau*
*Passau, Germany*

**Steffen Herbold**                                             *steffen.herbold@uni-passau.de*
*Faculty of Computer Science and Mathematics*
*University of Passau*
*Passau, Germany*

**Reviewed on OpenReview:** *https://openreview.net/forum?id=ddocn44Kaq*

## Abstract

Large Language Models (LLMs) are capable of answering many software related questions and supporting developers by generating code snippets. These capabilities originate from training on massive amounts of data from the Internet, including information from Stack Overflow. This raises the question whether answers to software related questions are simply memorized from the training data, which might raise problems as this often requires attribution (e.g., CC-BY license), sharing with a similar license (e.g., GPL licenses) or may even be prohibited (proprietary license). To study this, we compare responses to questions from Stack Overflow for questions that were known during LLM pre-training and questions that were not included in the pre-training data. We then calculate the overlap both with answers marked as accepted on Stack Overflow as well as other texts we can find on the internet. We further explore the impact of the popularity of programming languages, the complexity of the prompts used, and the randomization of the text generation process on the memorization of answers to Stack Overflow. We find that many generated answers are to some degree collages of memorized content and that this does not dependent on whether the questions were seen during training or not. However, many of the memorized snippets are common phrases or code and, therefore, not copyrightable. Still, we also have clear evidence that copyright violation happens and is likely when LLMs are used at large scales.

## 1 Introduction

A common criticism of generative Large Language Models (LLMs), such as ChatGPT,[1] Gemini,[2] and Claude[3] is that they are *stochastic parrots* (Bender et al., 2021). This means that they recombine existing texts based on inferred probabilities. Thus, under the stochastic parrot hypothesis, the LLMs generate collages

---

[1] https://chat.openai.com/
[2] https://gemini.google.com/
[3] https://www.anthropic.com/claude

in a probabilistic manner. When we consider the training procedures of such models, which are aimed at predicting the probability of the next token given an already existing text (*prompt*) (Radford et al., 2019) such a postulate makes sense. However, because the models learn probabilities of individual tokens, it is not clear if this really leads to a recombination of longer text fragments as well, if such fragments are re-used exactly as is, if they are modified (e.g., replacement of synonyms), or if the generated texts deviate more from the source training material. Moreover, while such a hypothesis is easier to argue for when the prompts are similar or equal to the training data, it is unclear what happens when we work with prompts that were not seen during training. Further, such models are not only trained to predict next tokens, but also post-trained to align their outputs with human preferences (Ouyang et al., 2022). To gain a better understanding regarding these aspects, our research is guided by the following research question: *To which degree are texts generated by LLMs collages of the training data?*

Within this study, we investigate this aspect of LLMs by considering data from the developer question and answer site Stack Overflow.[4] Thus, we study our research question with respect to data from the software engineering domain and for the task of answering open-ended questions. These answers generally require longer, semantically coherent text and expressions. Since Stack Overflow is part of many corpora for the training of LLMs (e.g. the Pile (Gao et al., 2020)), we assume that the question/answer pairs from Stack Overflow prior to the cutoff date for the creation of the training data are known. Vice versa, we can safely assume that question/answer pairs that were generated after the publication of the LLM are not used for training. We utilize this to create sample corpora of questions that were known and not known during training, along with their respective answers on Stack Overflow. Questions from the sample corpora are used as LLM-prompts. We then analyze the subsequence overlap of the LLM response with the true answers. In addition, we search for matching subsequences in a large, recent Web-Crawl (Granitzer et al. (2023)) to identify potential training data re-use outside of the Stack Overflow corpora. Studying these overlaps helps us to understand if generated texts are collages, where the data for the collages might have been sourced from, and if this effect is similar for questions seen during training and unknown questions. It also allows to study potentially arising copyright issues, particularly missing attributions. Our research will improve our understanding of how longer texts are related to the underlying training data, which has implications on the capabilities of an LLM to answer questions outside the scope of its training data as well as possible issues with copyright. This is not only relevant for the foundational research for the advancement of LLMs, but should also help us better understand the legal risks resulting from the capability of memorization that affect both text generation,[5] as well as image generation in bi-modal models.[6]

Our results indicate that memorization of common phrases and code is common and that outputs are typically, at least to some degree a collage of such memorized content, that is likely not protected by copyright. Importantly, this is the case for both questions seen during training and new questions. Only a small amount of memorized content was long enough and non-common, though we also found clear evidence of memorization that is likely violating copyright. This raises concerns when LLMs are used at large scales.

***Disclaimer:*** *This paper does not provide legal advice, but rather only a scientific analysis of data with respect to memorization and possible copyright concerns.*

## 2 Related Work

The previous work can be divided into three areas: plagiarism, privacy and security, and direct measurement of memorization. The first perspective on memorization is to consider plagiarism. Plagiarism is similar to memorizing content, especially when this is considered based on automated plagiarism checkers. Such plagiarism checkers are able to compare documents to an existing database of literature and determine passages that are textually equal or similar. de Wynter et al. (2023) use plagiarism detection to investigate LLM outputs regarding discourse and memorization of a diverse set of LLMs, e.g., ChatGPT, Bloom (Scao et al., 2023), and Galactica (Taylor et al., 2022). They find that up to 80% of outputs contain plagiarized content, but also that this can be reduced by adopting prompts that specifically tell the model to avoid

---

[4]https://stackoverflow.com/
[5]https://www.nytimes.com/2023/12/27/business/media/new-york-times-open-ai-microsoft-lawsuit.html
[6]https://spectrum.ieee.org/midjourney-copyright

memorized content. Lee et al. (2023) investigate types of plagiarism with pre-trained and fine-tuned models using GPT-2 (Radford et al., 2019) and different fine-tuned GPT versions, e.g., PatentGPT.[7] They find that GPT-2 plagiarized from OpenWebText (Gokaslan & Cohen, 2019) and that the fine-tuning has an impact on plagiarism of pre-training data: when the fine-tuning data overlaps with the pre-training data, this may reinforce memorization, while new fine-tuning data can reduce the memorization of pre-training data. Mueller et al. (2024) show that the number of potential copyright infringements varies greatly depending on the model, with bigger models generally producing more copyrighted texts.

In the second area, the authors come from a privacy and security perspective and investigate reproduction of private or security sensitive content. Huang et al. (2023) investigate reproduction of secret information in neural code models. They found that they can extract valid cloud credentials via autocomplete from Github Copilot[8] and Amazon CodeWhisperer.[9] Lukas et al. (2023) investigate the reproduction of Personal Identifiable Information (PII) from GPT-2. The authors propose attacks and mitigations for PII extraction and evaluate them on GPT-2. They show that GPT-2 can reveal PII, demonstrating the model's capability for memorization. More recently, Nasr et al. (2023) explore extraction of training data from aligned models via a novel divergence attack on open source models and ChatGPT. They show that they are able to extract training data at a high rate.

The final area is concerned with direct estimation of memorization in models and its impact on how the models work. Biderman et al. (2023) built a framework for comparing LLMs trained on the same data. The authors also evaluate memorization and find that a Poisson point process is a good approximation for the occurrence of memorized sequences over the process of training. Kandpal et al. (2022) explore the effect of duplication on memorization of training data. They show that duplicated data is memorized more effectively, demonstrating a need for deduplication of the training dataset as a measure to prevent memorization. Rabin et al. (2023) investigate the effects of noise and memorization in neural code models. They found that memorization is common in code models and that both correct but also noisy information is memorized. Carlini et al. (2023) quantify boundaries for the amount of memorization. The authors find that memorization scales log-linear with model size. This makes it important to study larger models as the authors largest model consists of 7B parameters. Similarly, Kiyomaru et al. (2024) show that memorization occurs more frequently with larger models, increasing prompt length and higher text frequency in the training data. However, they also discover that texts that are not part of later training steps are reproduced less often, even if they appear frequently in the training data. Wang et al. (2025) investigate how the task type affects memorization, showing that knowledge-intensive tasks exhibit the largest memorization effect. Schwarzschild et al. (2024) propose an algorithm for prompt-optimization which they use to assess memorization in the context of unlearning methods, demonstrating that models trained on the employed methods can still reproduce large parts of the supposedly unlearned data. While the above literature is focused on GPT-style models (Radford et al., 2019), Zeng et al. (2024) considered sequence-to-sequence models based on the T5 architecture (Raffel et al., 2020). They found that memorization is also common in sequence-to-sequence models, but also that multi-task fine-tuning reduces memorization.

While the prior work provides strong results regarding memorization, our work completes our understanding regarding two crucial, not yet studied issues. First, we specifically distinguish between memorization of training data, when we consider questions that were also part of the training and memorized content that is re-used for questions that are not directly observed during training. This allows us to understand to which degree memorization can happen in arbitrary tasks, even those that are not directly in the training. Second, we manually validate not only if memorization happens, but also if memorized fragments may be copyrightable. While some works, e.g., Mueller et al. (2024) did a similar analysis, this was on fairly long texts (at least 160 characters) and only for a task where they specifically asked to reproduce content. Instead, our analysis allows us to understand to which degree memorization occurs naturally in generated texts, as well as whether the memorization is rather just for common phrases and code versus actual protected content that has a certain level of originality.

---

[7] https://medium.com/artificial-corner/introducing-patentgpt-a-tool-for-extracting-technical-measurements-from-patents-80fa30c2bbd7

[8] https://github.com/features/copilot

[9] https://aws.amazon.com/de/codewhisperer/

### 2.1 Research Hypothesis

As described above, we want to understand the use of collages within generated texts as a means to study memorization. We use syntactic similarity observed through sequences of tokens that overlap between generated texts and reference as means to study collages. Further details on how this is measured are described later in Section 3.2. Based on the literature we discussed above, we derive the following concrete hypothesis that we want to test within this study.

**H1** LLM-generated answers to Stack Overflow questions are syntactically more similar to accepted human answers for questions observed during the pre-training of models in comparison to new questions.

**H2** LLM-generated answers for popular programming languages are syntactically further away from the training data than for niche languages.

**H3** Complex prompts following the style of questions in the training data increase the syntactic answer similarity to the training data and the amount of memorization within generated answers.

**H4** Generation of content without randomness increases that amount of memorized outputs.

Hypothesis **H1** is based on the common observation from the literature that LLMs are in general capable of memorization. Consequently, for out-of-distribution questions, i.e., questions not contained in the training data, the overlap of generated answers with answers in the training data should be smaller than for questions from the training. Importantly, if this prediction turns out to be false, this would in turn imply that all contents that LLMs generate are equally similar to the training data, regardless of whether the prompts were observed during training or not. Hypothesis **H2** is based on observations from the literature regarding the impact of duplication (Kandpal et al., 2022). These indicate that a larger diversity in data leads to lower memorization. Consequently, assuming that the diversity of the training data is higher for popular languages, the syntactic overlap with the training data should also be higher. Hypothesis **H3** is derived from the work by de Wynter et al. (2023), who found that the choice of a prompt impacts the amount of plagiarism and can be also used to reduce it. Vice versa, it should also be possible to use the prompts to trigger plagiarism, by requiring the LLMs to write in a certain style. Hypothesis **H4** follows from the technical aspects of the text generation process. If a response to a prompt is indeed memorized, this memorized content should have the highest probability given a prompt. The role of randomization of the next token based on the distribution from training is then to allow the model to deviate from the memorized content.

## 3 Methods

We now describe our research method in detail, i.e., the subjects, variables, how we execute the study to collect the required data, as well as the analysis of the collected data. Notably, we design our study protocol in a manner that does not require direct access to the training data. While concrete knowledge about the training process would be beneficial, this is unfortunately unrealistic for the recent generation of LLMs, where the training process has become a business secret. To still be able to study such models, we instead design our protocol around reasonable assumptions on the data used during the training process.

### 3.1 Subjects

The subjects of our study are two-fold: generative LLMs and Stack Overflow posts. The main subjects of our study are generative LLMs. We study these models by analyzing their outputs with respect to content from Stack Overflow.

To determine which LLMs we use as subjects, we define the inclusion criteria (see Table 4 in the appendix). The criteria have the goal to enable the evaluation of state-of-the-art models and at the same time enable us to guarantee that we can be reasonably sure that a) Stack Overflow posts were likely part of the training or b) Stack Overflow posts were not influenced by information provided by a LLM, posts where users did not use an LLM either in writing the question or for formulating an answer. Based on these criteria, we

| Date range | Popular languages | Niche languages |
|---|---|---|
| 01/2020-12/2020 | $P20$ with 157,558 question/answer pairs | $N20$ with 5,939 question/answer pairs |
| 09/2022-11/2022 | $P22$ with 32,148 question/answer pairs | $N22$ with 1,323 question/answer pairs |

Table 1: Identifiers and amount of posts retrieved for the different subsets of posts from Stack Overflow.

cannot consider the latest generation of models like ChatGPT 4, Gemini, or Llama 2 (Touvron et al., 2023) and newer, because the cutoff date for their training data is after the public release of ChatGPT. Overall, only two LLMs meet these criteria: ChatGPT 3.5 Turbo, [10] which is an instruction fine-tuned version of GPT-3 Brown et al. (2020) and Vicuna[11], which is an instruction-fine-tuned version of Llama Touvron et al. (2023). We decided to rather use the publicly available Vicuna model for our study, to facilitate future replication of our work. An additional concern with the use of GPT-3.5 Turbo would be updates that OpenAI has done since the release, that possibly extended the training data cutoff date. Such concerns do not exist with Vicuna.

Our second set of subjects are posts collected from Stack Overflow. We use the inclusion criteria depicted in Table 5. These inclusion criteria are designed to select pairs of questions and accepted answers in a manner that ensures a certain level of quality by requiring positive votes. Further, we collect only posts with certain tags to only include posts related to pre-defined programming languages. We use the Tiobe ranking of popularity of programming languages based on Web search data from 2020[12] to determine programming languages that are popular and those that are less popular, in the following referred to as popular and niche programming languages. This selection assumes that programming languages that are more often searched for by developers also receive more attention on Stack Overflow and that there are also more resources for such languages on the internet in general. We use the Top-10 programming languages from the Tiobe index as popular languages and ranks 51-100 to define niche programming languages. For each of these languages, we determine the related tags on Stack Overflow and then create subsets with all posts that contain one of the tags. Table 6 lists the tags we identified for this filtering. The selection of these subsets enables us to study the possible impact of the amount of information available on a topic on memorization.

For both subsets, we collect posts from 2020 as well as from June to November 2022. The data from 2020 was almost certainly part of the training of the models we used. The data from 2022 is too recent for the LLMs we study and, thus, was not used for their training of the foundation model, i.e., Llama. A risk that this data was used for the instruction tuning of the initial version of ChatGPT can also be ruled out, since this was done using hand-written responses by a dedicated team (Ouyang et al., 2022). The post-training of Vicuna has happened after the release of ChatGPT and is based on ChatGPT conversations collected in the ShareGPT[13] data.

Overall, this means we collect the four sets of Stack Overflow posts depicted in Table 1 and use these to evaluate how similar the answers to these questions generated by Vicuna are to the existing answers, as well as other information from the internet.

## 3.2 Variables

Since our goal is to understand to which degree the answers are collages from the training data, we define variables to measure the similarity between the generated answers by the LLMs and the other data sources. In this comparison, we are only interested in the syntactic similarity, not in the semantic similarity. Thus, we do not care about meaning, but rather only about the words and the combinations that were used. In terms of memorization, this means we only consider exact memorization. We conduct two such comparisons.

First, we compare the answer from the LLMs to the accepted answer given on Stack Overflow. To formally define this, we refer to a generated answer as $X^{gen} = (x_1^{gen}, ..., x_m^{gen})$ and to a reference answer from Stack

---

[10]https://platform.openai.com/docs/models/gpt-3-5-turbo
[11]https://lmsys.org/blog/2023-03-30-vicuna/
[12]https://web.archive.org/web/20201231062825/https://www.tiobe.com/tiobe-index/
[13]https://sharegpt.com/

Overflow as $X^{so} = (x_1^{so}, ..., x_{m'}^{so})$ such that $x_i^{gen}, i = 1, ..., m$ and $x_j^{so}, j = 1, ..., m'$ are the sequences of tokens of the generated answer/solution. Thus, we compare the token streams of the LLM answer and the Stack Overflow answer. We compare these token streams using metrics from the ROUGE suite for measuring text similarity Lin (2004). ROUGE metrics directly look at the overlap between subsequences between two texts, which is exactly what we are interested in with our syntactic comparison. A large overlap would be an indication for collages. Concretely, we use two variants of ROUGE. The first variant is *ROUGE-N*, which is defined as

$$ROUGE\text{-}N(n) = \frac{\sum_{gram_n \in X^{gen}} CountMatches(gram_n, X^{so})}{\sum_{gram_n \in X^{gen}} CountMatches(gram_n, X^{gen})} \quad (1)$$

where $gram_n \in X^{gen}$ are all $n$-grams from $X^{gen}$ and $CountMatches$ is a function that counts how often an $n$-gram match occurs in a sequence of tokens. Thus, we count how many of the $n$-grams from the generated texts appear in a reference answer from Stack Overflow.

Additionally, we compute

$$R_{lcs} = \frac{LCS(X^{gen}, X^{SO})}{m} \quad (2)$$

where $LCS$ computes the longest common subsequence between two texts. This metric is not directly available within ROUGE, but rather one of the components used to compute *ROUGE-L*, which would also use $m'$ to normalize the overlap, i.e., the length of the reference solution. However, since we are only interested in whether text from the reference is used in the generated answer and not if all text from the reference appears in the generated answer, using only $R_{lcs}$ gives more reliable data, in case the generated answers are shorter than the reference answers.

Since training data for LLMs is not limited to Stack Overflow, but is rather sourced from large Web crawls like Common Crawl,[14] it is also possible that the generated answers are collages of other texts from the training. While the training data is mostly based on data that can be searched for on the internet (with the possible exception of the book corpora used), the data set is neither fully specified nor publicly available. Consequently, we cannot search for overlaps between the generated answers and the training data directly. Instead, we simply use a Web search for exact matches to understand to which degree we have an overlap with data possibly seen during training. We take the pattern from *ROUGE-N* and define

$$WEB\text{-}ROUGE\text{-}N(n) = \frac{\sum_{gram_n \in X^{gen}} HasWebMatch(gram_n)}{|gram_n \in X^{gen}|} \quad (3)$$

where *HasWebMatch* is defined as whether there is an exact match for an n-gram with a Web search where we restrict the time to search hits older than the cutoff date for the pre-training. Specifically, we use 55 datasets from the OpenWebSearch project Granitzer et al. (2023)[15] from February to April 2024 for this purpose, summing up to a total size of 3.3 TB. Since we are interested in English Stack Overflow posts, we only considered the English part of the OWS data, roughly 1.3TB. Overall, the datasets contain crawls from 4.1M different hosts and 195M unique URLs. The data is publicly available under `https://openwebindex.eu`.

For *ROUGE-N(n)* and *WEB-ROUGE-N(n)*, we need to specify the length of the $n$-grams we want to consider for the memorization. The literature on plagiarism detection suggests four words for precise plagiarism detection based on $n$-grams (Barrón-Cedeño & Rosso, 2009). Considering the rule of thumb that a token is roughly $\frac{3}{4}$ of a word,[16] we should use $4 \cdot \frac{4}{3} = 5.33$ grams, which we round up to $n = 6$. Thus, we consider information to be memorized when we have six subsequent tokens overlapping between the generated sequence and the accepted answer for *ROUGE-N(n)*, respectively, the Web corpus for *WEB-ROUGE-N(n)*.

## 3.3 Data collection

We use a data dump from Stack Overflow[17] to collect the posts according to the criteria defined in Section 3.1. Once we have all data available, we generate answers for each question with Vicuna. We use two different

---

[14]`https://commoncrawl.org/`

[15]https://openwebsearch.eu/

[16]https://help.openai.com/en/articles/4936856-what-are-tokens-and-how-to-count-them

[17]https://archive.org/details/stackexchange

prompts to answer these questions (see Table 7 in the Appendix). With the simple prompt, we just ask the question directly without changing anything or providing any instructions to the LLM. This way, we get a "natural" reaction from the LLM that is not guided by us. With the second prompt, we try to bias the output in a direction, that makes a collage more likely, by first structuring the information in the same way it is available on Stack Overflow and then also asking for an answer in this style. The idea is that this prompt increases the likelihood that a generated answer is indeed a collage from Stack Overflow, because that is what we are actively looking for. We use two different temperatures to generate the answers. The temperature is a parameter that influences the randomness of the generated text by modifying the probability of sampling tokens. First, we set the temperature of the models to zero. This means that we avoid randomness and creativity in the outputs and just use the most likely next token all the time without sampling. Our rationale is that this increases the pressure on the model to create collages, because the randomness is part of what allows the model to generate new content. Next, we set the temperature to one. With a temperature of one, we sample according to the distribution of the next token determined by the LLM. This means that there is more randomness in the generation of the answers which could decrease the use of collages or mean that combined sentence fragments become shorter. Overall, this means we have 16 different sets of of results, i.e., the combinations of four subsets of Stack Overflow posts, i.e., $P20$, $N20$, $P22$, and $N22$; and four answers from the LLMs for each post with different prompts and temperatures. When presenting, analyzing, and discussing results, we identify these by a triple of year and programming language, prompt, and temperature, e.g., ($P20$, simple, low).

### 3.4 Statistical analysis

We compute the $R_{lcs}$, *ROUGE-N(n)* and *WEB-ROUGE-N(n)* for $n = 6$ as defined in Section 3.2 for all 16 data sets with different combinations of Stack Overflow posts, prompts, and temperature. For each approach, we report the mean value, as well as a 95% confidence interval for the mean value. We compute the confidence intervals in a non-parametric manner using bootstrap sampling Efron (1979). We use a Bonferroni-corrected significance level of $\alpha = \frac{0.05}{16 \cdot 3} \approx 0.001$ to compute these intervals to account for the multiple tests. We do not conduct additional statistical tests, but rather use the confidence intervals to determine significance. Hence, we conclude that results are significantly different with a confidence level of 95%, if their confidence intervals results do not overlap.

We analyze our hypothesis based on the results of our study and the expected observations in our data they predict.

- H1 predicts that the overlaps in text we compute for the data from 2020 should be larger than for the data from 2022, as the questions from 2020 were part of the training data. In our data, this means that we fix the prompt, and temperature, and then compare the P20 to the P22 data and the N20 to the N22 data. We expect that the overlap for P20/N20 is larger than for P22/N22.

- H2 predicts that the overlaps for niche languages are larger than for popular languages. In our data, this means that we fix the year, prompt, and temperature and compare the results for popular languages to niche languages. We expect that the overlap for N20/N22 is larger than for P20/P22.

- H3 predicts that a complex prompt can increase memorization. In our data, this means that when we fix the year, popularity, and temperature, we expect that the overlap is larger for the complex prompt than for the simple prompt.

- H4 predicts that less randomness means more memorization. In our data, this means that we fix the year, popularity, and prompt. We expect that the overlap for the low temperature is larger than for a higher temperature.

For all of the above, we carefully analyze the difference between $ROUGE-N(n)$ and $WEB-ROUGE-N(n)$ to understand from where information may have been memorized and to which degree generated language deviates from language on Stack Overflow and the internet in general.

### 3.5 Qualitative analysis of overlaps

While collages indicate that text fragments were memorized, this does not automatically imply that they would be copyrightable, as this requires a certain level of originality. For example, a common phrase like "our research is peer-reviewed" is sufficiently long to assume that this wording might have been memorized, but this would not be original and, therefore, copyrightable. To understand not only if we observe memorization, but also what is memorized, we conduct a qualitative analysis of the collages. Since our total corpus of data is too large for a full analysis over all languages, prompts, and years, we use sampling to collect a suitable subset for our qualitative analysis. We control for the length of the n-grams and the occurrence frequency during the sampling.

- The length of the n-gram. Short sequences with few tokens are less likely to meet the threshold for originality and are more likely to be randomly occurring. We consider short sequences at the threshold of what we consider collages (6-10 tokens), longer phrases (11-20 tokens) and longer texts (21+ tokens) as subgroups for the qualitative analysis.

- Occurrence frequency. How often a phrase is found in the Web corpus may be related to its originality, i.e., low frequency texts might be more original than high-frequency texts. However, high-frequency could still be original, e.g., the texts of license agreements. We distinguish if $n$-grams are found only once in the web corpus (unique), multiple times but less than the median in the corpus (2-4 times) and more often than the median (4 times).

For each combination of n-gram length and occurrence frequency, we sample 40 sequences, i.e., $3 \cdot 3 \cdot 40 = 360$ sequences of tokens that were generated by the LLM and found in the Web corpus.

Each sequence is independently annotated by two of the authors using deductive coding. The codes were defined as "common code", " common phrase", "copyrightable code", and "copyrightable phrase". Thus, the annotators were asked to determine if the n-gram is a code-snippet or some other phrase and to judge whether they believe this is original and, therefore, possibly copyrightable. Optionally, the annotators could provide a comment to explain their judgment. We report the confusion matrix of the agreements, as well as Cohen's $\kappa$ (Cohen, 1960) as measure for the reliability of the annotations.

## 4 Results

In the following, we present the results of our work. To facilitate replication, the data and code used to obtain these results are available online: `https://github.com/aieng-lab/replication-kit-stackoverflow-memorization/`

### 4.1 Overview of the metric distributions

Figure 1 shows the distribution of variables for each of the subgroups. The mean values, incl. confidence intervals can be found in Table 2. Our first observation is that for all plots, the difference between the data from 2020 and 2022 is very small. While we will look at this in greater detail later when studying the hypothesis, the broad trends regarding the syntactic overlaps are, apparently, *independent* from whether the data was seen during training.

In general, the 6-gram overlap between the actual answers and the generated answers measured with $ROUGE\text{-}N(6)$ is very low and, on average not more than 3.8% in any subgroup of our data. Similarly, answers are typically not fully memorized with a mean $R_{LCS}$ of at most 19% of the whole answer. However, looking at the distributions depicted in the violin plots, both distributions have long tails, indicating that while rare, stronger memorization does happen. When looking at the overlap of the generated answers and content found on the internet, we observe that the average $WEB\text{-}ROUGE\text{-}N(6)$ is between 10.6% and 17.5% for the respective subgroup. This is significantly higher than for $ROUGE(6)$, i.e., a notable part of the texts that are likely not random can be found as is on the internet. Whether this text was part of training is something we cannot answer due to a lack of access to the training data, but observing exact overlaps of six

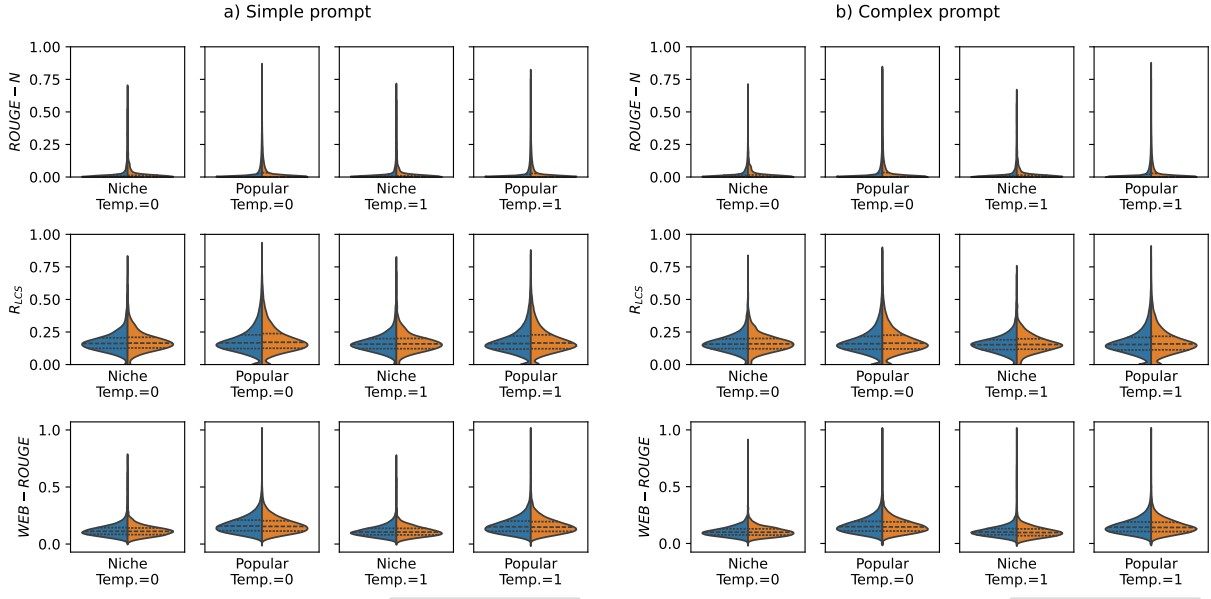

Figure 1: Distribution of the scores for all configurations. The violins are split by year.

| Set | Year | Complexity | Diff. | ROUGE-N(6) | | $R_{\mathrm{LCS}}$ | | WEB-ROUGE-N(6) | |
|-----|------|-----------|-------|------|------|------|------|------|------|
| | | | | **M** | **CI** | **M** | **CI** | **M** | **CI** |
| N | 20 | simple | low | 0.020 | [0.018, 0.022] | 0.173 | [0.170, 0.175] | 0.119 | [0.117, 0.121] |
| | | simple | high | 0.018 | [0.016, 0.020] | 0.167 | [0.165, 0.170] | 0.114 | [0.111, 0.116] |
| | | complex | low | 0.020 | [0.018, 0.022] | 0.165 | [0.162, 0.168] | 0.106 | [0.104, 0.107] |
| | | complex | high | 0.018 | [0.016, 0.020] | 0.158 | [0.155, 0.161] | 0.107 | [0.106, 0.109] |
| | 22 | simple | low | 0.021 | [0.017, 0.025] | 0.176 | [0.170, 0.182] | 0.117 | [0.112, 0.121] |
| | | simple | high | 0.019 | [0.015, 0.023] | 0.168 | [0.162, 0.174] | 0.112 | [0.108, 0.116] |
| | | complex | low | 0.020 | [0.017, 0.024] | 0.166 | [0.161, 0.172] | 0.105 | [0.101, 0.109] |
| | | complex | high | 0.020 | [0.016, 0.024] | 0.162 | [0.156, 0.167] | 0.102 | [0.098, 0.106] |
| P | 20 | simple | low | 0.034 | [0.033, 0.034] | 0.186 | [0.185, 0.187] | 0.171 | [0.170, 0.171] |
| | | simple | high | 0.030 | [0.030, 0.031] | 0.179 | [0.179, 0.180] | 0.164 | [0.164, 0.165] |
| | | complex | low | 0.033 | [0.032, 0.033] | 0.177 | [0.176, 0.177] | 0.159 | [0.158, 0.159] |
| | | complex | high | 0.029 | [0.029, 0.030] | 0.169 | [0.168, 0.170] | 0.154 | [0.154, 0.155] |
| | 22 | simple | low | 0.038 | [0.037, 0.039] | 0.192 | [0.191, 0.195] | 0.166 | [0.165, 0.167] |
| | | simple | high | 0.034 | [0.033, 0.036] | 0.185 | [0.184, 0.187] | 0.161 | [0.159, 0.162] |
| | | complex | low | 0.036 | [0.035, 0.038] | 0.182 | [0.181, 0.184] | 0.156 | [0.155, 0.157] |
| | | complex | high | 0.033 | [0.032, 0.034] | 0.174 | [0.172, 0.176] | 0.152 | [0.151, 0.153] |

Table 2: Mean values (M) and confidence intervals (CI) for each metric and each subgroup within our data.

tokens with that ratio is unlikely to be an artifact of our evaluation. Still, this is a strong indication that generated texts are, at least partially, indeed collages of the training data.

Exact memorization of answers is rare but seems to happen in data seen during pre-training. Interestingly, we also observe very similar syntactic memorization patterns for answers not seen during pre-training. Furthermore, all generated texts have a notable overlap with text available on the internet, indicating that answers are also, at least partly, borrowed from other sources.

|     | Comparison | $ROUGE\text{-}N(6)$ | $R_{LCS}$ | $WEB\text{-}ROUGE\text{-}N(6)$ |
|-----|------------|---------------------|-----------|--------------------------------|
| **H1** | (P20, simple, low) vs. (P22, simple, low) | -0.004 | -0.006 | -0.005 |
|     | (P20, simple, high) vs. (P22, simple, high) | -0.004 | -0.006 | -0.003 |
|     | (P20, complex, low) vs. (P22, complex, low) | -0.003 | -0.005 | - |
|     | (P20, complex, high) vs. (P22, complex, high) | -0.004 | -0.005 | -0.002 |
|     | (N20, simple, low) vs. (N22, simple, low) | - | - | - |
|     | (N20, simple, high) vs. (N22, simple, high) | - | - | - |
|     | (N20, complex, low) vs. (N22, complex, low) | - | - | - |
|     | (N20, complex, high) vs. (N22, complex, high) | - | - | - |
| **H2** | (N20, simple, low) vs. (P20, simple, low) | -0.014 | -0.013 | -0.062 |
|     | (N20, simple, high) vs. (P20, simple, high) | -0.012 | -0.012 | -0.050 |
|     | (N20, complex, low) vs. (P20, complex, low) | -0.013 | -0.012 | -0.053 |
|     | (N20, complex, high) vs. (P20, complex, high) | -0.009 | -0.011 | -0.047 |
|     | (N22, simple, low) vs. (P22, simple, low) | -0.017 | -0.016 | -0.049 |
|     | (N22, simple, high) vs. (P22, simple, high) | -0.015 | -0.017 | -0.049 |
|     | (N22, complex, low) vs. (P22, complex, low) | -0.016 | -0.016 | -0.051 |
|     | (N22, complex, high) vs. (P22, complex, high) | -0.013 | -0.012 | -0.050 |
| **H3** | (P20, simple, low) vs. (P20, complex, low) | - | +0.009 | +0.012 |
|     | (P22, simple, low) vs. (P22, complex, low) | - | +0.010 | +0.010 |
|     | (N20, simple, low) vs. (N20, complex, low) | - | +0.008 | +0.013 |
|     | (N22, simple, low) vs. (N22, complex, low) | - | - | +0.012 |
|     | (P20, simple, high) vs. (P20, complex, high) | - | +0.010 | +0.010 |
|     | (P22, simple, high) vs. (P22, complex, high) | - | +0.011 | +0.009 |
|     | (N20, simple, high) vs. (N20, complex, high) | - | +0.009 | +0.007 |
|     | (N22, simple, high) vs. (N22, complex, high) | - | - | +0.010 |
| **H4** | (P20, simple, low) vs. (P20, simple, high) | +0.004 | +0.007 | +0.007 |
|     | (P22, simple, low) vs. (P22, simple, high) | +0.004 | +0.007 | +0.005 |
|     | (N20, simple, low) vs. (N20, simple, high) | - | - | +0.005 |
|     | (N22, simple, low) vs. (N22, simple, high) | - | - | - |
|     | (P20, complex, low) vs. (P20, complex, high) | +0.004 | +0.008 | +0.005 |
|     | (P22, complex, low) vs. (P22, complex, high) | +0.003 | +0.008 | +0.004 |
|     | (N20, complex, low) vs. (N20, complex, high) | - | +0.007 | - |
|     | (N22, complex, low) vs. (N22, complex, high) | - | - | - |

Table 3: Difference of mean values between subgroups. Results are only reported if the confidence intervals, as reported in in Table 2, are not overlapping. The groups are arranged such that the hypothesis predicts that the first-mentioned group has a higher mean value which should yield a positive value. Negative values are contradicting our hypothesis.

## 4.2 Evaluation of Hypothesis

In the following, we will look at these result with respect to our hypothesis in detail. This analysis is based on Table 3 that summarizes the differences between relevant subgroups for the hypotheses H1-H4.

**H1: LLM-generated answers to Stack Overflow questions are syntactically more similar to accepted human answers for questions observed during the pre-training of models in comparison to new questions.** Our results indicate that there are significant differences for popular programming languages. For the niche languages, we do not observe any significant differences. However, contrary to our expectation, the direction is that the overlap between the generated and actual answers is *higher* for the P22 data than for the P20 data, i.e., the generated answers are more similar to answers not seen during training. This is consistent across all three metrics.

> We reject H1 and even observe a slight opposite trend indicating that the newer, unseen data is syntactically closer to the LLM output than data that was part of the pre-training.

**H2: LLM-generated answers for popular programming languages are syntactically further away from the training data than for niche languages.** Our results provide a strong indication for the opposite, i.e., that the memorization of content for popular languages is significantly higher than for niche languages. This is consistent across all subpopulations and all three metrics. For *ROUGE-N*(6), the difference is between 0.9% and 1.7%. While these numbers seem small, recall that the highest average we observed for *ROUGE-N*(6) was only 3.8%, which makes the difference a statistically large effect, although low in absolute terms. For $R_{LCS}$, we observe differences between 1.1% and 1.7%. While consistent, the effect is a lot weaker than for *ROUGE-N*(6), because the average of $R_{LCS}$ are higher. The strongest difference can be observed for *WEB-ROUGE*(6), where the increase is between 4.7% and 6.2%.

> We reject H1 and observe the opposite, i.e., that memorization is more common for popular languages, which indicates that data diversity does not hinder memorization. We hypothesize that this might mean that topic prevalence is the dominant factor for memorization, i.e., topics that are frequently occuring are more likely to be memorized, regardless of the data diversity.

**H3: Complex prompts following the style of questions in the training data increase the syntactic answer similarity to the training data and the amount of memorization within generated answers.** Our results provide a weak signal that this might be the case. While we observe no significant difference in *ROUGE-N*(6), we see small increases in $R_{LCS}$ between 0.8% and 1.1% for all data subpopulations, except those with the N22 data. *WEB-ROUGE-N*(6) also slightly increases with values between 0.7% and 1.3%. This indicates that the complex prompt slightly shifts the distribution of the output to create syntactically slightly longer common subsequences with the actual answers, but also to use more *n*-grams that can also be found on the internet in general. Both happen without significantly increasing the overlap of the *n*-grams with the actual answer.

> Our results for this hypothesis are inconclusive. We find hints towards complex prompts increasing the ratio of memorized outputs, but the signal is weak and inconsistent across metrics.

**H4: Generation of content without randomness increases the amount of memorized outputs.** Our results show a consistent, but small increase in memorization for the popular languages. For niche languages, this can only be observed for $R_{LCS}$ with the N20 data and the complex prompt. When we combine this data with the insights from H2 that show that memorization is stronger for popular languages, we find that our hypothesis is partially confirmed.

> We accept our hypothesis for the popular language for which we find a consistent but small increase in all metrics, and reject it for the niche languages. We hypothesize that this means that low randomness can only increase memorization that is sufficiently strong, which only applies to the popular languages in our data, based on our results for H2.

## 4.3 Qualitative insights into memorized data

For this analysis, we only considered data for which we observed the largest overlap with the actual answers based on the results for the hypotheses above. Thus, we selected data from popular projects (H2) from 2022 (H1) and a low temperature (H4). We decided to label data for both the simple and the complex prompt, because the results for H3 are inconclusive. According to the commonly used criteria by McHugh (2012), the agreement between both raters is moderate, i.e., $\kappa \in [0.6, 0.8]$. A detailed look at the disagreements revealed that this is driven by different interpretations of corner case regarding what constitutes code. Rater A typically judged corner cases (e.g., Markdown table headers) as phrases, rater B judged such fragments as code.

The fragments we find as is on the internet are, from our perspective, mostly not copyrightable: we observed aspects like import statements, standard layout parameters, calls to functions with generic names

and parameters, or phrase like "follow these steps". However, we also found six data points for which one of the raters deemed that this might be sufficient for copyright protection (see Table 9 in the appendix). We scrutinized these data points further to find their possible sources and understand if they indeed violate copyright. Two samples seem to be strong candidates for copyright violations. The first is a definition that is quoted from Wikipedia. The second is the text of an error message from the Spring framework. Definitions are protected by copyright,[18] for error messages it depends on the use case, e.g., when reporting an error it is fair use, however, when designing own products re-use may be a violation.[19] Another snippet is an API call from a product documentation, which means that this likely does not violate copyright and would be fair use.[20] Two other snippets are part of the formatting information, once of a table, once of a Web site. Such formatting information is typically not copyrightable.[21] Finally, we also have one case, where the text we considered memorized from training data was instead only copied from the question, i.e., a false positive for memorization. Overall, the most conservative reading of these findings is that there is only on clear copyright violation out of 720 samples we checked.

Assuming a user uses an LLM daily only once to answer a question, this would mean that there would roughly one copyright violation every two years. While this number seems very low, LLMs may be used at very large scales. Assume you have a mid-sized company with 100 employees, working 220 days a year, querying an LLM on average four times a day. This would mean there are $\frac{220 \cdot 100 \cdot 4}{720} \approx 122$ copyright violations per year.

> Overall, few possibly memorized texts lead to possible copyright violations, meaning that individuals who sometimes use LLMs have a low risk. However, power users and companies almost certainly, at least sometimes, generate memorized content that could be protected by copyright.

## 5 Discussion

Our results confirm prior work that LLMs may memorize contents, even with instruction tuning. While we did not study the last generation of LLMs, but rather the first generation that used instruction tuning, we are not aware of any fundamental differences in the training procedure that would imply that our findings do not hold, i.e., that memorization can still happen. A similarly controlled experiment is, unfortunately, not possible with newer LLMs, because we cannot ensure that data is neither influenced by LLMs (may be been generated) or was used for their training.

We believe that the crucial finding from our work is not only that there is still memorization, which is hardly surprising, nor that this can sometimes lead to outputs that are protected by copyright. Instead, we believe that the crucial aspect that our work demonstrates is that this happens regardless of whether the prompt was part of the training data. This is also a key difference to prior work on memorization, e.g., by Carlini et al. (2023) who only demonstrated that the completion of texts that are from the training data may be memorized or Mueller et al. (2024) who directly ask to generated specific content. In contrast, we specifically tested with prompts that were not part of the training, i.e., questions asked on Stack Overflow after pre-training was finished. Still, the answers for these questions had often overlap with texts from the internet and our analysis shows that at least one answer violates copyright.

Moreover, our results indicate that, at least for the Q&A scenario we studied, the degree to which outputs are memorized does not depend on whether the question was part of pre-training. Indeed, the overlap between the actual answers and the generated answers was even higher for questions not seen during training. This strongly hints at the fact that LLMs always generate collages to some degree, regardless of whether the prompts are known from training or not.

---

[18]See, e.g., `https://law.stackexchange.com/questions/102147/can-copy-pasting-a-word-definition-from-a-dictionary-site-cause-a-copyright-issu`

[19]See, e.g., `https://law.stackexchange.com/questions/40864/are-computer-generated-error-messages-subject-to-copyright`

[20]see, e.g., `https://law.stackexchange.com/questions/73565/using-software-api-documentation-without-copyright-infringement`

[21]see, e.g., `https://aeonlaw.com/copyright-protection-css-and-html`

# 6 Limitations

A clear limitation of our work is that Vicuna, an LLM several years old, was our only study subject. As discussed above, we chose Vicuna because we wanted a controlled setting in which we knew there was data that had not been seen during training and had not been influenced by users who posted LLM-generated content. Consequently, we cannot guarantee that our results will translate to newer model generations. While it is reasonable to assume that the results will hold for models with a similar architecture and prompting strategy, it is highly uncertain for multi-hop prompting and reasoning models Ji et al. (2025), which have a slightly different output generation strategy involving the production of a stream of thinking tokens. Despite the limitations of translating the results to newer architectures, our study paves the way for future research because we found that memorization in generated content does not depend on whether the questions were seen during training. Thus, future studies do not require such strong control over when the data was seen during training, enabling research on newer models. Instead, the focus should be on determining whether supposedly memorized output was generated by an LLM, which would also reduce legal uncertainties in LLM usage.

The second clear limitation of our work is that we found only a single clear case of a copyright violation and one more candidate that depends on the context. This threatens the generalizability of our claim that power users and companies almost certainly, at least sometimes generate memorized content. However, we also found three more memorized data points in our sample that are sufficiently unique to be copyrightable, they were just from contexts that either allow re-use due to fair-use exemptions of copyright or for textual data that is in general not copyrightable. Thus, while the cases in which we actually found copyrightable content were lower, the cases where we found relatively unique, uncommon statements were more with five hits. Therefore, while we cannot rule out that the cases were we found copyrightable content were outliers and that they are a lot more uncommon than our data indicates, the existence of other cases that are just not copyrightable because they fall under exceptions of copyright laws, indicate that this is unlikely.

# 7 Conclusion

We conducted a controlled experiment in which we studied to which degree generated answers are the same as actual answers, for both questions seen and unseen during training. We selected our data in such a way that it is unlikely that any actual answer is AI generated, by selecting a time-frame prior to the release of ChatGPT. We found that whether questions were seen or not during the training has no strong impact on the syntactic similarity between the actual and generated answers. In contrast, syntactic similarity was even slightly higher for unseen questions. We also observed that questions for common topics are more likely to lead to memorization and that randomization in the text generation processes decreases memorization. For all questions, we found significant overlap of 6-grams with texts found on the internet, hinting at possible memorization and manually studied this for 720 possibly memorized snippets. While these were mostly harmless, e.g., common code like import statements, we also found one clear copyright violation and one candidate for a copyright violation. Notably, the copyright violation happened in a context where we did not try to trigger it, e.g., by giving a prompt from a protected source and asking for completion. Instead, it happened for a Stack Overflow question unseen during training by quoting Wikipedia without reference. Thus, while most queries to LLMs are likely not violating copyright, even these rare cases add up to a possibly massive number of copyright violations, considering the scale at which LLMs are used.

## Broader Impact Statement

The copyright implications of LLM use have huge societal implications and the question if and to which degree LLMs violate copyright is already the topic of multiple lawsuits. Our work provides an indication that while generated answers are to some degree collages from the training data, copyright violations are rare for individual cases but common at scale, independent of whether the prompts are known from the training.

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

# A Appendix

## A.1 Additional details for the data collection

Within this appendix, we provide additional details with respect to our inclusion criteria (Table 4 and 5), the tags used to identify the popular and niche programming languages (Table 6, and the prompts we used for generation answers (Table 7).

| Inclusion criteria for generative LLMs | Rationale |
|---|---|
| Instruction fine-tuning was used | The latest generation of LLMs, starting with Instruct-GPT (Ouyang et al., 2022) clearly outperforms older LLMs. |
| Pre-training data older than December 2022 | ChatGPT was released to the public on Nov 30th, 2022. We need to ensure that we have a period of time, in which ChatGPT was not available yet and where the data was also not used to pre-train the LLM. |
| Instruction fine-tuning data publicly available or older than December 2022 | We need to be able to ensure that the fine-tuning data is not related to the Stack Overflow posts we study. |

Table 4: Inclusion criteria for the selection of suitable LLMs for our study.

| Inclusion criteria for Stack Overflow posts | Rationale |
|---|---|
| Has accepted answer | Posts without answer are not suitable for our study because we cannot compare the answers to the output of the LLMs. That the answer is accepted is in indicator for the correctness of the answer. |
| Has a positive vote for the question | Posts without a positive vote may be unclear questions, which would not be suitable for our use case. |
| Has a positive vote for the answer | Posts without a positive vote may be unclear answers, which would not be suitable for our use case. |
| Has no duplicates and is not a duplicate | Posts with duplicates may introduce noise in our analysis. |

Table 5: Inclusion criteria for suitable Stack Overflow posts for our study.

| Subset criteria for Stack Overflow posts | Rationale |
|---|---|
| Tagged as <c>, <java>, <python>, <c++>, <c#>, <vb6>, <vb6.net>, <javascript>, <php>, <r>, <sql> | Popular programming language with huge communities and a vast number of publicly available resources beyond Stack Overflow. |
| Tagged as <actionscript>, <applescript>, <autolisp>, <awk>, <bash>, <bc>, <bourne-shell>, <control-language>, <clojure>, <coffeescript>, <common-lisp>, <elixir>, <elm>, <elisp>, <emacs-lisp>, <erlang>, <f#>, <factor-lang>, <forth>, <genie>, <hacklang>, <haskell>, <icon-language>, <inform7>, <iolanguage>, <j>, <kornshell>, <ladder-logic>, <maple>, <mercury>, <mql4>, <ocaml>, <opencl>, <openedge>, <oz>, <pl-i>, <q-lang>, <raku>, <rexx>, <rpg>, <smalltalk>, <spark-ada>, <spss>, <stata>, <vala>, <vbscript>, <verilog> | Niche technology with a limited amount of publicly available resources. |

Table 6: Tags used to identify subsets of popular and niche programming languages based on the ranking of programming languages in the Tiobe Index 2020.

| Prompt ID | Prompt |
|---|---|
| Simple | [QUESTION] |
| Complex | The following question comes from Stack Overflow and has the following structure:
Title: [TITLE OF THE QUESTION]
Tags: [TAGS OF THE QUESTION]
Question: [QUESTION]

Provide an answer in the style of Stack Overflow. |

Table 7: Prompts used to generate answers for the Stack Overflow questions using the LLMs.

### A.2 Details about the qualitative analysis

Figure 2 reports the confusion matrices of the qualitative labeling conducted by two authors to understand what the memorized text fragments are. Table 8 lists examples for outputs that were memorized, but are common phrases or code and, therefore, not copyrightable. Table 9 lists the candidates for copyright violations that we found, including their likely sources, what kind of content we have, and whether this is possibly protected by copyright.

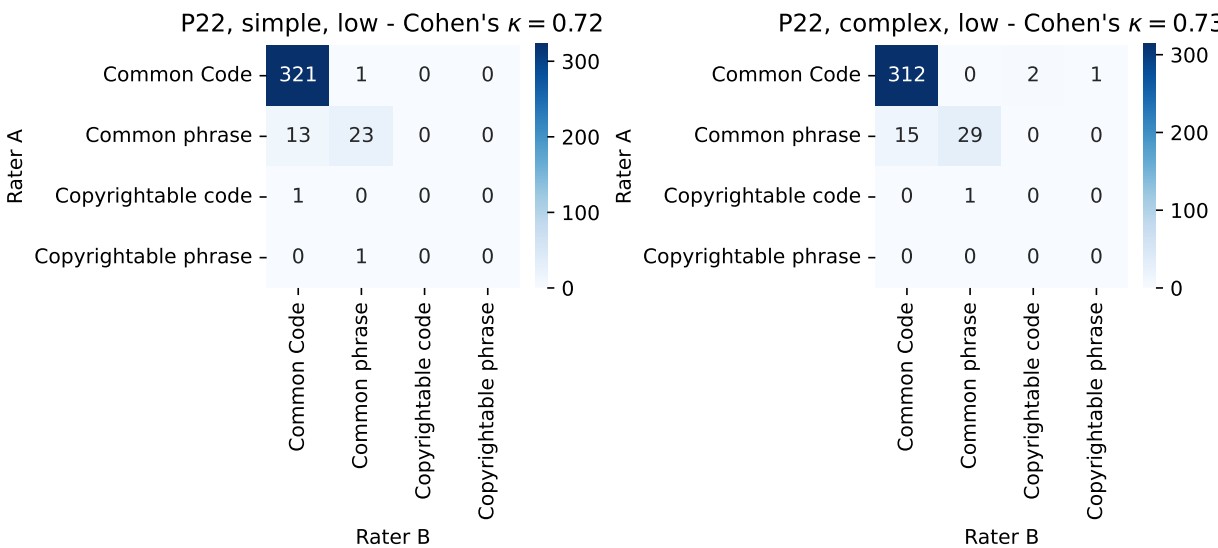

Figure 2: Confusion matrices for the qualitative analysis that compare the judgment of both raters.

| Memorized output |
|---|
| .amazon.awssdk.http.apache |
| pd.Series([0, 1, 2, 2, 3, np |
| '); for (var i = 0; i < checkboxes.length; i++) { if (checkboxes[i].name |
| d on your description, it seems like you're experiencing issues with Du |
| If you have any further questions or need additional assistance, please don't hesitate to ask. Good luck with your |
| This can lead to a significant increase in the size of the workflow object |

Table 8: Examples for memorized outputs that are likely not copyrightable because they are common code (first three examples) and common phrases (bottom three examples).

| Memorized output | Likely source | Type |
|---|---|---|
| a mechanical device for applying pressure to an inked surface rest | Multiple, e.g., `https://en.wikipedia.org/wiki/Printing_press` | Definition. Protected. |
| `expected at least 1 bean which qualifies as autowire candidate.` | Many. Common error message of the Spring framework. | Error message. Protected for some purposes. |
| `do_action ('woocommerce_before_single_ product_summary'` | `https://wp-kama.com/plugin/woocommerce/hook/woocommerce_before_single_product_summary` | API Call. Fair use. |
| `-------- --------` `+\|id\|select_type\|table\|partitions\|type\|` `possible_keys\|key\|key_len\|ref\|rows\|` `filtered\|Extra\|+` | Unknown | Table header. Typically not copyrightable. |
| `container {display: grid;` `grid-template-columns: repeat(4,1fr);` `grid-gap: 10px; background-color: #21` | Unknown | Format. Typically not copyrightable. |
| `class; /** * Define the model's default state. * * @return array */ public function definition() {return [ 'name' => $this->faker->words(3, 7)` | Text was also part of the question. No memorization. | - |

Table 9: Possibly memorized model outputs deemed as potentially copyrightable by one of the authors.

