# OpenReview forum: "Studying memorization of large language models using answers to Stack Overflow questions"
_TMLR — Accepted by TMLR_

### Review · Reviewer_mrvh · 2025-07-27

**Summary Of Contributions:**

The core contribution of this paper is its in-depth study of memorization in LLMs, specifically examining whether generated answers to programming questions are collages of existing content from their training data.The paper investigates further by comparing LLM responses with StackOverflow questions that were known during pre-training versus unknown future questions.

The author specifically differentiates memorization of questions seen during training from 2020 and the content re-used for questions not directly observed during training from 2022. This is a crucial distinction from prior papers, which often focused on completing texts directly from training data or asking for specific content reproduction. The paper finds that the phenomenon of generating collages occurs regardless of whether the questions were seen during training or not. In fact for popular programming languages, LLM output were even more similar to unseen data.

The study used Vicuna and ChatGPT 3.5 Turbo model, and here are some key findings:
- Topic Popularity has a dominant factor in memorization
- Low randomness with temperature=0 increase the memorization scores.
- Complex Prompts Show Inconclusive Signal.
- Low syntactic overlap, but high web overlap: While the ROUGE-N(6) overlap with accepted Stack Overflow answers was very low (at most 3.8%), the WEB-ROUGE-N(6) showed significantly higher overlap (10.6% to 17.5%) with general internet content. This indicates that a notable portion of generated texts can be found as-is on the internet, strongly suggesting that generated texts are, at least partially, collages of training data
- Copyright violations: Most memorized snippets were not copyrightable, but the study also provided clear evidence that copyright violations do occur rarely.

Key Strengths:
- Addresses a critical & timely issue: The paper is very relevant in the current LLM world, to understand its memorization & its implication for copyright violations. The research would contribute to a lot of future lawsuits.
- The study focuses on seen vs unseen data which provides a better understanding on memorization factors in LLMs and whether memorization even occurs with arbitrary tasks not directly part of the training data.
- Very systematic & controlled experiments.
- Copyright validations: the analysis manually validates whether memorization fragments are actually copyrightable , moves beyond mere text overlaps to access originality.
- Key insights on other factors that could influence memorization. example the finding on how popular programming languages have higher memorization suggests topic prevalences play a dominant role with LLMs.

Weaknesses:
- The study is based on older LLM models, primarily using Vicuna (fine tuned Llama) & ChatGPT 3.5. The newer might could have a very different results, but the authors is making assumption that there wouldn't be an foundational differences from those older models.
- The weak signals for complex prompts in (H3) states as "inconclusive" with weak signal. Most corporate use-cases of agenetic AI or advanced automations use complex prompts and it would be interesting to have a bit more finding and provide some conclusive direction with advanced prompts.
- The evidence of Copyright violation from 720 manually checked sample is a relatively small qualitative sample for the usage of today's LLMs.

**Audience:**

Yes

**Audience Explanation:**

TMLR audience would be interested in this papers finding. The paper directly investigates a critical aspect of Large Language Models (LLMs) specifically their memorization behaviors and copyright violations.

This understanding of "possible issues with copyright" is crucial for both "foundational research for the advancement of LLMs" and for comprehending "legal risks". Example the copyright violation low occurrence is highlighted on the paper, which makes it feel like it would have less impact for individuals utilizing the LLM vs a corporate with large number of employees, which the violation occurrence would be amplified.

**Broader Impact Concerns:**

This paper includes a detailed "Broader Impact Statement" that highlights the societal implications of copyright violation, demonstrating its relevance to ethical concerns within the machine learning community.

**Claims And Evidence:**

Yes

**Claims Explanation:**

The paper provides very strong and clear evidence for its key contributions and claims, particularly within the scope of the LLMs and data studied.
 - Very clear evidence on in-depth study of Memorization & distinguishing seen vs. unseen data set. The controlled experiment is very well designed.
- They quantify memorization using syntactic similarity metrics like ROUGE-N(6), RLCS, and WEB-ROUGE-N(6). They are valid measure of similarity, specifically focusing on the overlap of 6-word sequences but they do have their limitation as well, which might go beyond the scope of this paper.
- They have gone deeper in identifying the influencing factoring for memorization, example- with temperature setting, popular languages, etc.

**Requested Changes:**

I'm overall supportive of the paper, but here is something to strengthen it further:
- Run the same or similar experiment finding with the newer models which would make this paper very relevant, as the old model usages are drastically different from the current usage. The newer model do use some of the real-time data in their response, so it would be interesting to see how the perform in terms on memorization & copyright concerns.
- Include Illustrative Examples of "Collage" Snippets. Beyond the focus on potentially copyrightable snippets, the paper could include a small table examples of the results about the discussed sections.

---

> ### Author Response · Authors · 2025-08-08
>
> Thank you for your effort reviewing our paper!
>
> ## Weaknesses
>
> > The study is based on older LLM models, primarily using Vicuna (fine tuned Llama) & ChatGPT 3.5. The newer might could have a very different results, but the authors is making assumption that there wouldn't be an foundational differences from those older models.
> > The evidence of Copyright violation from 720 manually checked sample is a relatively small qualitative sample for the usage of today's LLMs.
>
> We added a new section 6 on Limitations, that discusses these aspects more prominently.
>
> > The weak signals for complex prompts in (H3) states as "inconclusive" with weak signal. Most corporate use-cases of agenetic AI or advanced automations use complex prompts and it would be interesting to have a bit more finding and provide some conclusive direction with advanced prompts.
>
> While we agree with the sentiment, a confirmative study design as used here can easily degenerate into p-hacking when deciding to dig deeper for results that are not yet conclusive. Therefore, we did not add additional analysis within this paper but believe that future studies should be set up in a way that they put a strong focus on measuring this, including in agentic settings.
>
>
>
> ## Requested changes
>
> > Run the same or similar experiment finding with the newer models which would make this paper very relevant, as the old model usages are drastically different from the current usage. The newer model do use some of the real-time data in their response, so it would be interesting to see how the perform in terms on memorization & copyright concerns.
>
> We agree that this is a very relevant experiment. However, the scope of this paper is specifically chosen so that we are able to distinguish between memorization in outputs for questions seen during training and completely new questions. Adding experiments that break with this requires additional considerations, e.g., which (and how many) models to choose, determining if the same questions from Stack Overflow are suitable, etc. Consequently, we leave this to future work.
>
> > Include Illustrative Examples of "Collage" Snippets. Beyond the focus on potentially copyrightable snippets, the paper could include a small table examples of the results about the discussed sections.
>
> We extended Appendix A.2 with examples with examples for common code and phrase (new Table 8). For space reasons, we decided to keep the examples exclusively in the Appendix.

---

> > ### Comment · Reviewer_mrvh · 2025-09-03
> > **Acknowledging of the Authors' Revision**
> >
> > Thanks for addressing some of the suggestions. Adding limitation section addresses the scope of the paper and calls out the potential for the future work in this space. Good with expansion of Appendix A.2 with examples for common code and phrase .
> >
> > I still believe TMLR audience would be interested in this papers findings and it could provoke some thought for some similar work with the latest LLM models.

---

### Review · Reviewer_wwa9 · 2025-08-04

**Summary Of Contributions:**

This paper studies the level of memorization in LLMs under different scenarios. It uses 3 different models, opensource (Vicura, LLAMA) and close sourced (GPT-3.5-Turbo). It uses software engineering related questions from Stack Overflow as the subject set of study, and splits the questions and answers into seen/unseen sets in the models' pre-training. Since the exact pre training data is not available to the authors, a rough time-based split is used. The paper uses ROUGE (for n-gram overlapping) as the metrics for memorization, supplemented by longest common string (LCS) and qualitative human review.

The study proposed 4 questions, each contains a factor that may affect the level of memorization. Experiments showed that the level of memorization is low and not likely to be affected by most of the factors. The authors also perform close-up human inspection on the longer sequences that could be from memorization, to see if they are likely to be a concern from copyright perspective. Since there's only a few examples that are truly copyright-worthy memorization amount ~700 examples, this is likely a non-zero but minor concern.

Strength.

- Well written, with clear hypotheses and explanations on possible points of concern.
- Ask important and interesting questions.
- Large scale experiments with significance estimation.
- Careful human inspection of the results and qualitative analysis.

**Additional Comments:**

none

**Audience:**

Yes

**Audience Explanation:**

Memorization of LLM under different scenario is an interesting and important topic.

**Broader Impact Concerns:**

We might want to be careful about claiming LLMs impact on copyright infringement only based on very small samples.

**Claims And Evidence:**

No

**Claims Explanation:**

- Made too many weak assumptions, for example

(1) GPT-3.5-Turbo won't see newer data. Instruction tuning, or even tail-patching with new data is a common practice in flag-ship models. It's risky to assume the pre-training cutoff date is the firewall between old/new data.
(2) WEB-ROUGE is an approximation of training data. The pre-training data is usually carefully ablated and cleaned, with possible format adjustment. The differences from web search could be non-trivial.
(3) Single-digit possible copyright data cases may not be a reliable estimation on how much this could occur on larger scale data; the sample is too small.

- The definition of "simple" vs "complex" questions is not clear. If I understand correctly, "complex" means the question is asked in the similar format of Stack Overflow? How would this make the question more complex?
- Under-estimation of noise / variance. The paper uses mean/variance from the samples to get the confidence interval, however there are also several layers of relatively weak assumption before the random sampling part, the actual confidence-interval that's meaningful to the rejection of the hypothesis could be higher.

**Requested Changes:**

Reviewer understands that the more concerning weakness is harder to address given the experiment is done, and the lack of access to true pre-training data, but for open source models we should have the true pre-training set. It's much more convincing if wo do a white-box experiment with clear access to the pre-training dataset.

The conclusion about copyright and "weak holds" of the hypothesis feels less supported to the reviewer. Maybe we can tune down the certainty?

The Discussion and Conclusion sections have some commonality, maybe it's better to combine them?

Minor typos: First paragraph of Related Work, "fine-tunded" -> "fine-tuned".

---

> ### Author Response · Authors · 2025-08-08
>
> Thank you for your work reviewing our paper!
>
> ## Weaknesses
>
> > GPT-3.5-Turbo won't see newer data. Instruction tuning, or even tail-patching with new data is a common practice in flag-ship models. It's risky to assume the pre-training cutoff date is the firewall between old/new data.
>
> While we agree there could be such issues with GPT-3.5-Turbo, we can rule out with reasonable certainty that there was such an effect with the Vicuna model we studied. We know with certainty that there was no knowledge update of any kind of the underlying Llama model, nor was there a second round of instruction tuning after the release of Vicuna. These considerations further support our model choice and we updated the manuscript in Section 3.1 as follows: *An additional concern with the use of GPT-3.5 Turbo would be updates that OpenAI has done since the release, that possibly extended the training data cutoff date. Such concerns do not exist with Vicuna.*
>
> > WEB-ROUGE is an approximation of training data. [...]
>
> First, even if data is further processed for training, finding longer, exact text matches that are not memorized is highly unlikely and could not explain the high number of overall matches we find. Due to that, we do not believe that the matches we find are not due to memorization. Since modifications and format adjustment would decrease the memorizations from the training data we could find on the Web, if anything, our results rather underestimate the actual memorization rates.
>
> Second, we want to stress that is has, unfortunately, become unrealistic for all recent major models to have direct access to training data and, therefore, be able to conduct white-box studies on these models: even though some companies still release open-weights models, training code and data are not shared. Therefore, we believe that this is rather a strength of our method and a core contribution of this paper: a method that can study memorization for models, for which we do not have access to the training data.
>
> > Single-digit possible copyright data cases may not be a reliable estimation [...]
>
> We agree with this concern and discuss this in a newly added Section 6 ("Limitations").
>
> > The definition of "simple" vs "complex" questions is not clear. [...]
>
> "Simple" and "complex" do not refer to the difficulty of the questions, but rather to the style of the prompt. The symple prompt just naivly asked the question directly, while the complex prompt provides information about the question ("comes from Strack Overflow"), the structure of the question including title, tags, and the question itself, and the request that the answer is in the style of Stack Overflow. The intent is to observe if this biases the model towards more direct memorization from Stack Overflow and enable measurements for H3.
>
> > Under-estimation of noise / variance. The paper uses mean/variance from the samples to get the confidence interval, however there are also several layers of relatively weak assumption before the random sampling part, the actual confidence-interval that's meaningful to the rejection of the hypothesis could be higher.
>
> We are not sure which assumptions are meant here. Instead of assuming that the data follows a normal distribution and computing the confidence intervals based on the mean, variance, and the z-value, we rather use a non-parametric method to estimate the confidence intervals in a robust manner without assumptions on the data. The method we employ is based on bootstrap sampling (Efron, 1979) and uses the central limit theorem: we simply resample with replacement from our data and compute for each of these samples the mean value. This gives us a distribution of the expected mean value, from which we can then determine the confidence interval by taking the 2.5-percentile and the 97.5 percentile of this distribution. For a better explanation of this method, we refer to this short blog post: http://acclab.github.io/bootstrap-confidence-intervals.html
>
>
>
> ## Requested changes
>
> > Reviewer understands that the more concerning weakness is harder to address given the experiment is done, and the lack of access to true pre-training data, but for open source models we should have the true pre-training set. [...]
>
> See above.
>
> > The conclusion about copyright and "weak holds" of the hypothesis feels less supported to the reviewer. Maybe we can tune down the certainty?
>
> We added the limitations section to highlight the limitations of the generalizability of our results.
>
> > The Discussion and Conclusion sections have some commonality, maybe it's better to combine them?
>
> The partial overlaps in content between the conclusion and other sections are intentional, since many cursory readers of papers focus only on the abstract, introduction, and conclusion. Therefore, these sections are structured such that they contain the key aspects of our study, even though this leads to redundancies.
>
> > Minor typos: [...]
>
> Fixed

---

> > ### Comment · Reviewer_wwa9 · 2025-08-24
> >
> > Thanks for the answers!
> >
> > "GPT-3.5-Turbo won't see newer data". Agreed with the author's discussion. Adding the clarification would help.
> >
> > "WEB-ROUGE is an approximation of training data."  Again I understand the difficulty of getting the pre-training data, but this is still a limit from a research accuracy perspective. Adding clarification would help.
> >
> > "Single-digit possible copyright data cases may not be a reliable estimation". Clarification helps, but this is still a limit and weakens the paper's claim.
> >
> > "We agree with this concern and discuss this in a newly added Section 6". Thanks! This sections looks good.
> >
> > "The definition of "simple" vs "complex" questions is not clear." Now my question is answered. What about we name them as "vanilla question format" vs "StackOverflow question format"? complex could easily lead to misunderstanding as difficulty.
> >
> > "Under-estimation of noise / variance." My question is fully answered. Thanks!
> >
> > "The Discussion and Conclusion sections have some commonality" I see the rationale now, however duplicated information is still less preferred.

---

### Review · Reviewer_LzfT · 2025-08-06

**Summary Of Contributions:**

This paper makes three key contributions to understanding LLM memorization and copyright risks. Methodologically, it introduces a novel framework for studying memorization without requiring access to proprietary training data, using temporal boundaries to distinguish seen vs. unseen content. Empirically, it reveals the surprising finding that LLMs memorize content equally whether questions were in training data or not, suggesting they always generate "collages" regardless of prompt familiarity. Practically, it quantifies copyright violation risk: while individual violations are rare (~1 in 720 cases), the massive scale of LLM usage means copyright infringement is "likely when LLMs are used at large scales," providing crucial evidence for ongoing legal debates about AI and intellectual property.

**Audience:**

Yes

**Audience Explanation:**

This paper offers groundbreaking insights into LLM memorization with significant implications across multiple research domains. Its **novel methodology** enables studying memorization without accessing proprietary training data, making it broadly applicable. The **surprising finding** that memorization occurs equally for training-seen and unseen questions fundamentally challenges assumptions about how LLMs generate text, suggesting they always create "collages" regardless of prompt familiarity. This has profound implications for **AI safety, interpretability, and copyright research**. The work provides empirical evidence for ongoing legal debates while offering a framework for future studies. Its intersection of technical AI research with practical societal concerns makes it highly relevant to the broader research community.

**Broader Impact Concerns:**

The paper should address more concerns in the broader impact statement, including a discussion of potential impacts on:

  - Stack Overflow community and attribution norms
  - Educational implications for programming learning
  - Economic effects on content creators
  - Potential mitigation strategies
  - Ethical considerations for AI companies

**Claims And Evidence:**

No

**Claims Explanation:**

The paper exhibits some **scope-claim misalignment**:

**Broad "LLM" generalizations vs. single model testing:**
- Claims: *"Large Language Models (LLMs) are capable of..."* and *"LLMs always generate collages to some degree, regardless of whether the prompts are known from training or not"*
- Experiments: *"We decided to rather use the publicly available Vicuna model for our study"*

**Precise copyright violation extrapolations from tiny samples:**
- Claims: *"Assume you have a mid-sized company with 100 employees...This would mean there are 220·100·4/720 ≈ 122 copyright violations per year"*
- Experiments: Only 360 manually coded samples, with just *"one clear copyright violation and one candidate for a copyright violation"*

**Universal memorization claims:**
- Claims: *"This strongly hints at the fact that LLMs always generate collages to some degree"*
- Experiments: Despite testing only one model under limited conditions

**Requested Changes:**

Fix or explain the scope-claim misalignment

---

> ### Author Response · Authors · 2025-08-08
>
> Thank you for your effort reviewing our paper!
>
> > The paper exhibits some scope-claim misalignment:
> > Broad "LLM" generalizations vs. single model testing:
> > Claims: "Large Language Models (LLMs) are capable of..." and "LLMs always generate collages to some degree, regardless of whether the prompts are known from training or not"
> > Experiments: "We decided to rather use the publicly available Vicuna model for our study"
> >
> >Precise copyright violation extrapolations from tiny samples:
> > Claims: "Assume you have a mid-sized company with 100 employees...This would mean there are 220·100·4/720 ≈ 122 copyright violations per year"
> > Experiments: Only 360 manually coded samples, with just "one clear copyright violation and one candidate for a copyright violation"
> >
> > Universal memorization claims:
> > Claims: "This strongly hints at the fact that LLMs always generate collages to some degree"
> > Experiments: Despite testing only one model under limited conditions
>
> We agree that the scope of our study affects our claims and that we should discuss this more prominently. Therefore, we added a new Section 6 that discusses and highlights these limitations.

---

### Author Response · Authors · 2025-08-08
**Response to Reviews**

We want to thank all reviewers for their constructive and timely reviews. We updated our paper based on your comments. The largest change is the addition a new limitations section (Section 6).

Please note that we did not update the broader impact statement due to conflicting statements in the reviews: one says it should be broader, one says it should be more careful, and one review seems to like it the way it is. Consequently, we opted for the latter and did not modify this at this time.

---

### Decision · Action_Editor_L5Hd · 2025-09-08

**Recommendation:** Accept as is

**Audience:**

Yes

**Audience Explanation:**

LLM memorization is a timely and important topic for the TMLR community. While, as previously noted, the submission suffers from several limitations and presents relatively modest conclusions, most reviewers, myself included, believe it represents a meaningful first step toward addressing this issue. It has the potential to spark further exploration and contribute to future developments in this area.

**Claims And Evidence:**

Yes

**Claims Explanation:**

This paper investigates memorization in LLMs, specifically highlighting that responses to programming questions are collages of content from training data. By comparing LLM outputs on Stack Overflow questions known during training (from 2020) with previously unseen ones (from 2022), the authors show that collage-like behavior frequently occurs, regardless of whether the question was part of the training set. In particular, for popular programming languages, overlap with unseen data is even more pronounced. A key strength of the paper is its systematic comparison of seen versus unseen data, along with manual validation. However, the study has several limitations, including its reliance on older models, inconclusive results for complex prompts, relatively weak evaluation metrics, and a small-scale analysis of potential copyright violations.